# Simulation Analysis of Knee Ligaments in the Landing Phase of Freestyle Skiing Aerial

**Yanming Fu [1,2], Xin Wang [3,*] and Tianbiao Yu [1,*]**

[1]  School of mechanical engineering and automation, Northeastern University, Shenyang 110819, China
[2]  Laboratory management center, Shenyang Sport University, Shenyang 110102, China
[3]  School of kinesiology, Shenyang Sport University, Shenyang 110102, China
[*]  Correspondence: wangxin@syty.edu.cn (X.W.); tianbiaoyudx@gmail.com (T.Y.)

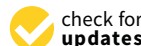

**Featured Application: This paper focused on the knee joints of freestyle skiers in three landing conditions (neutral, backward, or forward landing). Aim to understanding the force inside the knee joints during the landing phase. The research results will be used in the design of athletes' protective gear.**

**Abstract:** The risk of knee injuries in freestyle skiing athletes that perform aerials is high. The internal stresses in the knee joints of these athletes cannot easily be directly measured. In order to ascertain the mechanical response of knee joints during the landing phase, and to explore the mechanism of damage to the cartilage and ligaments, a finite element model of the knee joint was established. Three successful landing conditions (neutral, backward, or forward landing) from a triple kicker were analyzed. The results demonstrate that the risk of cruciate ligament damage during a neutral landing was lowest. A forward landing carried medium risk, while backward landing was of highest risk. Backward and forward landing carried risk of injury to the anterior cruciate ligament (ACL) and posterior cruciate ligament (PCL), respectively. The magnitude of stress on the meniscus and cartilage varied for all three landing scenarios. Stress was largest during neutral landing and least in backward landing, while forward landing resulted in a medium level of stress. The results also provide the basis for training that is scientifically robust so as to reduce the risk of injury and assist in the development of a professional knee joint protector.

**Keywords:** freestyle skiing aerials; knee joint; ligament; finite element simulation

## 1. Introduction

In the absence of trauma, the knee joint can operate effectively for decades while being subjected to high mechanical loads. The knees of athletes have a shorter life expectancy. After professional athletes retire, their knee joints often exhibit damage due to overwork. For example, the rate of knee joint injury in freestyle skiing aerialists in Chinese national team is close to 85% and higher in retired athletes. Studies have shown that instantaneous impact in the vertical direction will damage the cartilage of the knee joint, while long-term repeated impact will cause strain damage to the stress concentration region of the cartilage [1]. Other studies have shown that when the knee joint flexes at a certain angle, shear or torsion stress caused by instantaneous movement of the tibia may damage the cruciate ligaments [2,3], while instantaneous inversion or eversion of the knee may cause the medial or lateral collateral ligaments to be damaged [4,5]. In addition, the athletes' ankle joints are essentially locked in snowshoes, which do not provide sufficient cushioning at the moment of landing, resulting in greater impact force to be absorbed by the knee joints. For these reasons, freestyle skiing aerialists suffer a high rate of knee injuries.

In order to establish the internal force within the knee joint, a number of researchers have conducted in vitro experiments and with cadavers [6,7]. Other researchers have used the inverse dynamics technique to analyze knee movement [8,9]. Most researchers choose finite element analysis [10], which is possibly a more objective method of obtaining the specific movements of components within the knee joint. In this study, finite element simulation method had been adopted to analyze the knee joint at phase of landing, so as to obtain the stress of cartilage and ligament during landing buffering.

## 2. Methods

### 2.1. 3D Reconstruction

In the early stages of the study, CT (Computed Tomography) and MR (Magnetic Resonance) test data were obtained from a male athlete volunteer who had provided signed informed consent [11]. The height and weight of the volunteer was close to the mean value of male athletes. The CT and MR data were imported into Mimics software. After 3D reconstruction, the optimized model was imported into Abaqus software to complete finite element analysis.

Bone tissue 3D reconstruction was derived from CT data, while cartilage and ligament tissue models were derived from MR data [12–14]. Bone, cartilage, and ligament tissues were reconstructed independently using Mimics. The models were imported into Geomagic software in STL format. After image registration and model assembly, the bone, cartilage and ligament models were finally stored independently in IGES format.

### 2.2. Mechanical Models

From the motions observed during landing, a mechanical model of the landing phase was established in the X and Y directions, as shown in Figure 1. The angle between the femur and tibia was defined as α. The angle between the tibia and the slope of the landing plane was defined as β. The angle between the trunk and the landing slope was defined as γ. $G$ and $G'$ represented force due to gravity and force on the tibial plateau, respectively. By combining theory with research [15,16] $G'$ was calculated as:

$$G' = 85.6\% \times G$$

The proportions of the force on the medial and lateral femoral condyles have been estimated to be 60% and 40%, respectively [17]. Therefore, if air resistance and wind direction are neglected, the impact force on the tibial plateau can be calculated from an analysis of the projectile motion of the human body.

### 2.3. Kinematic Analysis

Velocity of takeoff was obtained from a camera located at the kicker and the velocity of center of gravity, left and right knee joint movement angle, and duration of the balancing phase of landing at the point of impact were collected from cameras located on both sides of the landing site on the landing slope. Data relating to the kinematic parameters described above were obtained through SIMI Motion analysis. The maximum height of the trajectory above the landing site could be calculated from the takeoff velocity of a freestyle skiing aerialist completing a triple somersault having a prescribed action prior to landing (slope of 37.5°). This was approximately 17 m, as shown in Figure 1.

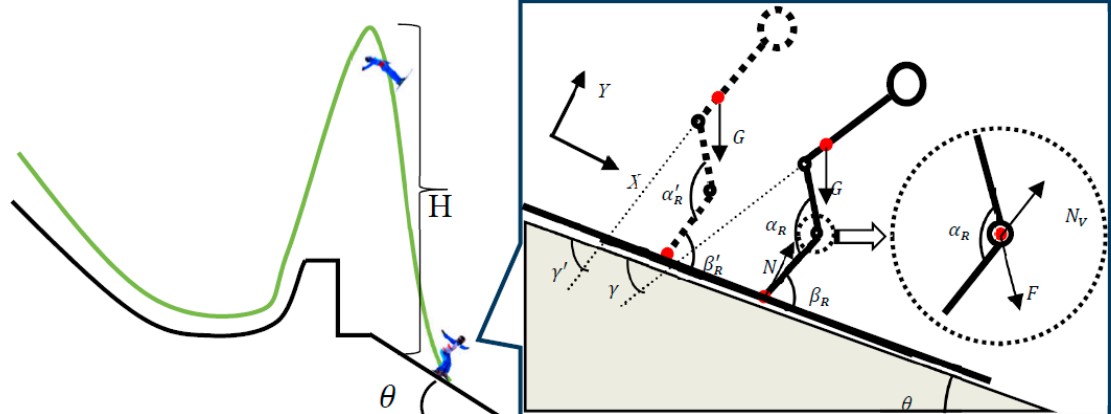

**Figure 1.** Freestyle skiing aerial motion curve and landing phase of the kinematic analysis diagram ($\alpha'_R$ and $\beta'_R$ represent the angles of the right knee and right ankle as landing begins, respectively, $\gamma'$ was the angle between the torso and the landing slope. $\alpha_R$ and $\beta_R$ represent the angles of the right knee and right ankle at the end of the balancing phase of landing, $\gamma$ represents the angle between the torso and the landing slope at the end of the balancing phase of landing. $H$ was the distance from the highest position of the trajectory to the landing position. $\theta$ was the landing slope angle. $G$ was the force experienced by the athlete due to gravity, $N$ was the reaction force from the landing slope, $N_v$ was the reaction force from the tibial plateau, $F$ was the impact force from the upper part of the knee joint in the direction of the femur).

In the process of image acquisition, 150 video files and 50 groups of images of the motion were collected in this study. After screening, 30 groups of images met the experimental requirements. After statistical analysis, no significant difference was detected between the experimental data and that of the mechanical model established in previous studies. The data collected were combined with the results calculated from the mechanical model and used as boundary conditions in the finite element calculation.

## 2.4. Establishing the FEM (Finite Element Model)

The 3D models obtained by 3D reconstruction were imported into Abaqus software in IGES format with the results of meshing (Figure 2a, using C3D10 meshing classification), attributes of materials and the assembled model displayed in Figure 2. The finite element calculations were completed using constraint conditions which were also configured in Abaqus, and then the density, elastic modulus and Poisson's ratio of bone, cartilage, and meniscus were set according to the data in Figure 2b [18–21]. An incompressible Neo-Hooke hyperelastic model was selected for the ligament model, expressed by C10 and D1 coefficients, as shown in Figure 2c [22]. The constraint conditions of the knee FEM were set according to the anatomical characteristics. All the contacts between ligaments and bones were "Tie" type constraint and tibia was set as "ENCASTRE".

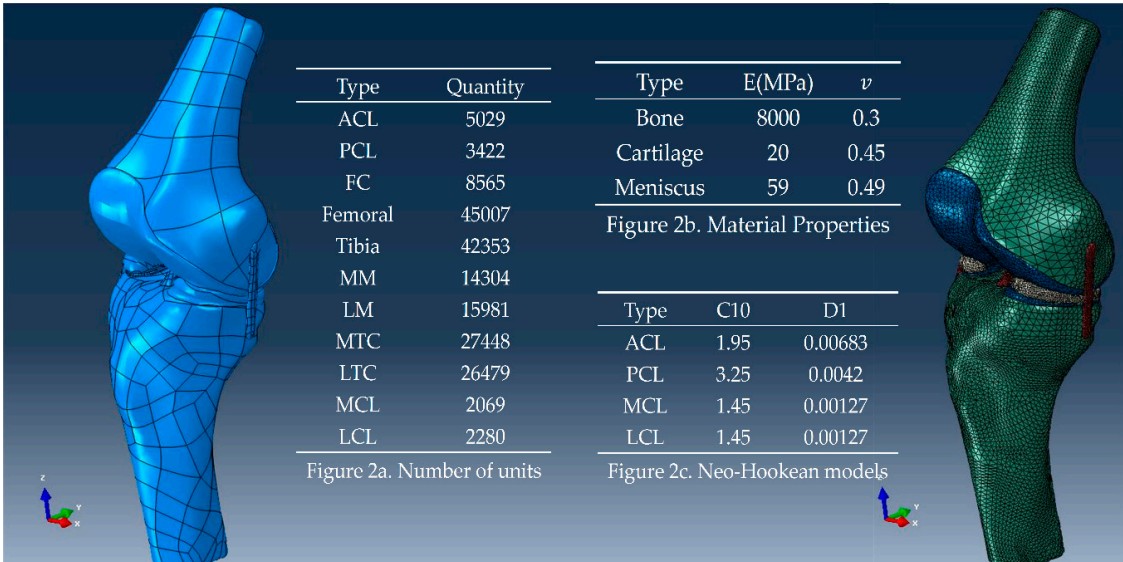

| Type | Quantity |
|---|---|
| ACL | 5029 |
| PCL | 3422 |
| FC | 8565 |
| Femoral | 45007 |
| Tibia | 42353 |
| MM | 14304 |
| LM | 15981 |
| MTC | 27448 |
| LTC | 26479 |
| MCL | 2069 |
| LCL | 2280 |

Figure 2a. Number of units

| Type | E(MPa) | $v$ |
|---|---|---|
| Bone | 8000 | 0.3 |
| Cartilage | 20 | 0.45 |
| Meniscus | 59 | 0.49 |

Figure 2b. Material Properties

| Type | C10 | D1 |
|---|---|---|
| ACL | 1.95 | 0.00683 |
| PCL | 3.25 | 0.0042 |
| MCL | 1.45 | 0.00127 |
| LCL | 1.45 | 0.00127 |

Figure 2c. Neo-Hookean models

**Figure 2.** Mesh generation and material properties of knee models. The right figure represents assembled knee model. In the left figure, different colors define the different material properties. ACL: anterior cruciate ligament; PCL: posterior cruciate ligament; FC: femoral cartilage; MM: medial meniscus; LM: lateral meniscus; MTC: medial tibial cartilage; LTC: lateral tibial cartilage; MCL: medial collateral ligament; LCL: lateral collateral ligament.

## 3. Results

### 3.1. Results of Kinematic Analysis

Using SIMI Motion, 30 groups of motions were parsed. Mean velocity ($\overline{v}$) and standard deviation (*SD*) at takeoff were calculated as: $\overline{v} \pm SD = 16.62 \pm 2.51$ (m/s). In addition, the mean duration of the balancing phase ($\overline{t}$) and *SD* of landing were measured as: $\overline{t} \pm SD = 0.16 \pm 0.04$. The values of $\alpha_L$, $\alpha_R$, $\beta_L$, $\beta_R$, and $\gamma$ for the landing balancing phase are shown in Figure 3.

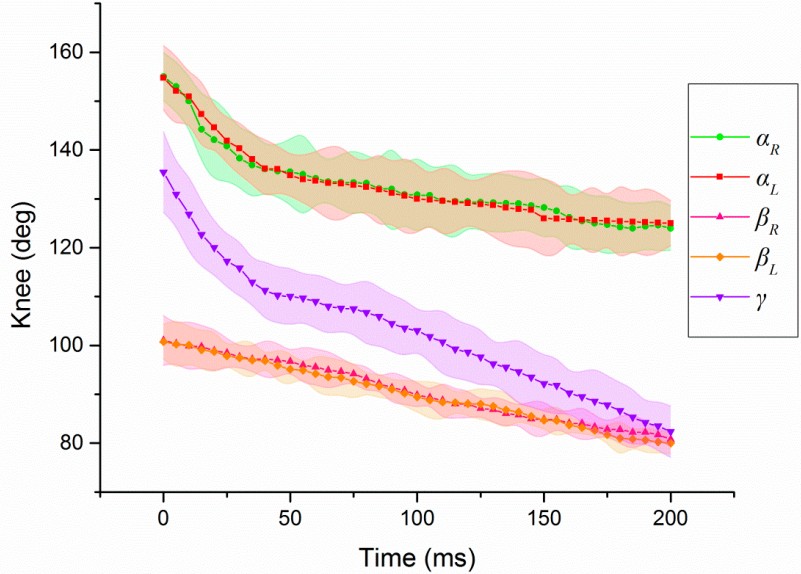

**Figure 3.** Variation trend of human body angle during the balancing stage, where $\alpha_L$ represents the angle of the left knee, $\alpha_R$ the angle of right knee, $\beta_L$ the angle of left ankle. $\beta_R$ the angle of right ankle. $\gamma$ represents the angle between the torso and landing slope.

From videos of successful landings, a successful landing can be approximately divided into three variations, as shown in Figure 4. The first can be denoted forward landing due to the front of the skis touching the snow first with the athlete's center of gravity being in a forward orientation. This requires an aerialist to adjust their center of gravity backward to ensure a successful landing. The second is denoted neutral landing, where the front and rear of the skis touch the snow simultaneously with the athlete's center of gravity mid-way of all the variations. This requires maintenance of knee joint stability and center of gravity to ensure a successful landing. The third is described as a backward landing, where the rear of the skis touch the snow first with the center of gravity of the athlete rearward. This requires an adjustment of the center of gravity forward to ensure a successful landing.

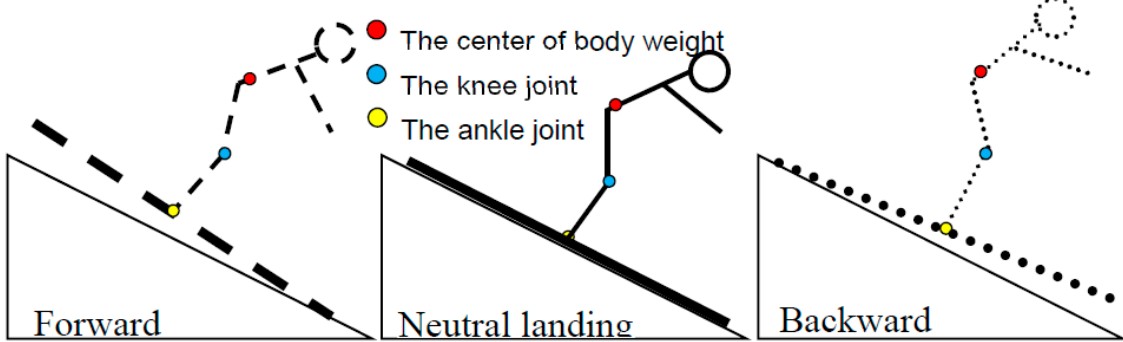

**Figure 4.** The three variations of successful landing.

More apparent differences can be observed if the three are combined into one image. An integrated image can be obtained when the center points of the knee and ankle joints from the three are overlain, as shown in Figure 5. Vertical lines from the center of body weight demonstrate the different positional features. In forward landing, this line is forward of the knee joint, essentially located within the boundaries of the knee joint in neutral landing and behind the knee joint in backward landing.

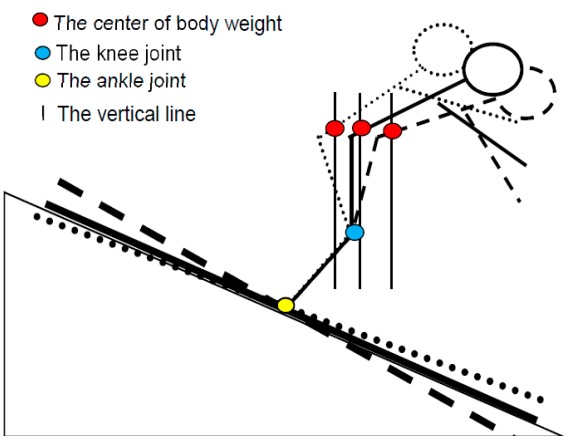

**Figure 5.** Overlay effect.

### 3.2. Results of Mechanical Model Analysis

$\alpha$, $\beta$, and $\gamma$ are the principal parameters for the mechanical model, with takeoff velocity ($v$) and landing balancing duration ($t$) critical variables for dynamic analysis. By adding results of tests using a force platform in the laboratory [23], vertical and shear force curves of the landing slope and tibial plateau were fitted by dynamic analysis, as shown in Figure 6. The balancing phase was 0–200 ms, maximum vertical force $F_\sigma$ and shear force $F_\tau$ on a single knee tibial plateau were approximately 2527 and 1699, respectively, with the range of knee angles from 155 to 120 degrees. Peak force was experienced at approximately 135°.

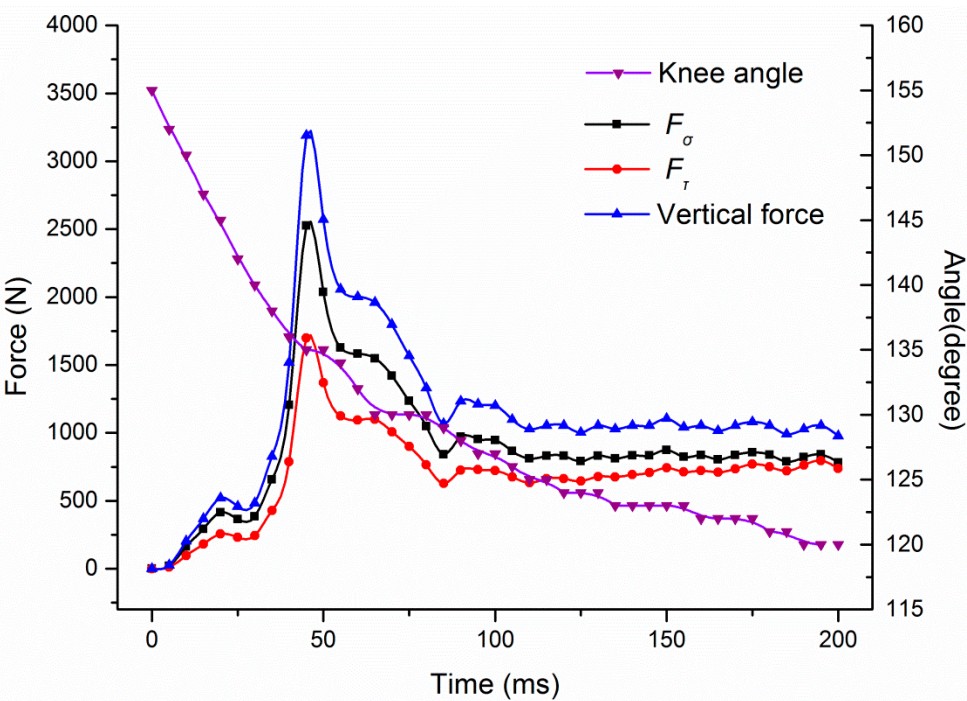

**Figure 6.** Force curve of a single knee joint in the balancing phase.

### 3.3. Finite Element Simulation Results

Combined with the three landing conditions described in Section 3.1, the external constraint conditions of the model were adjusted and finite element analysis performed on the ACL, PCL, MCL, and LCL. Time–stress curves were calculated according to the results of the analysis.

In forward landing (Figure 7), deformation of the front of the skis was minimal with elastic force in the balancing stage that could be ignored. According to the results of the analysis, due to the tendency of the tibia to move rearwards, the position of stress concentration was at the lower end of the posterior bundle of the ACL with a maximum stress value of 4.01 MPa. The position of stress concentration on the PCL was at the upper end of anterior bundle, with a maximum value of 6.81 MPa. The position of stress concentration on the LCL was in the middle, with a maximum stress value of 1.60 MPa. Maximum stress on the MCL was 1.96 MPa, with stress concentration in the middle and on the lower section. As the center of gravity was adjusted rearwards, the medial meniscus became squeezed, resulting in a region of stress concentration in the middle of the MCL.

In neutral landing (Figure 8), the skis were parallel to the landing slope and so elastic force in the balancing process was ignored. Although the impulse from the femur had a forward component at the tibial plateau, the tibial plateau also moved synchronously with flexion of the knee joint and so relative motion of the femur and tibial plateau were not apparent. According to the results of the finite element analysis, the position of stress concentration was at the lower end of posterior bundle of the ACL with a maximum stress of 3.16 MPa. Stress was concentrated in the upper part of the posterior bundle of the PCL to the lower end of anterior bundle with a maximum stress of 6.61 MPa. Maximum stress on the MCL was 2.49 MPa which was concentrated in the middle region. Stress concentration was in the middle of the LCL, to a maximum stress value of 1.67 MPa.

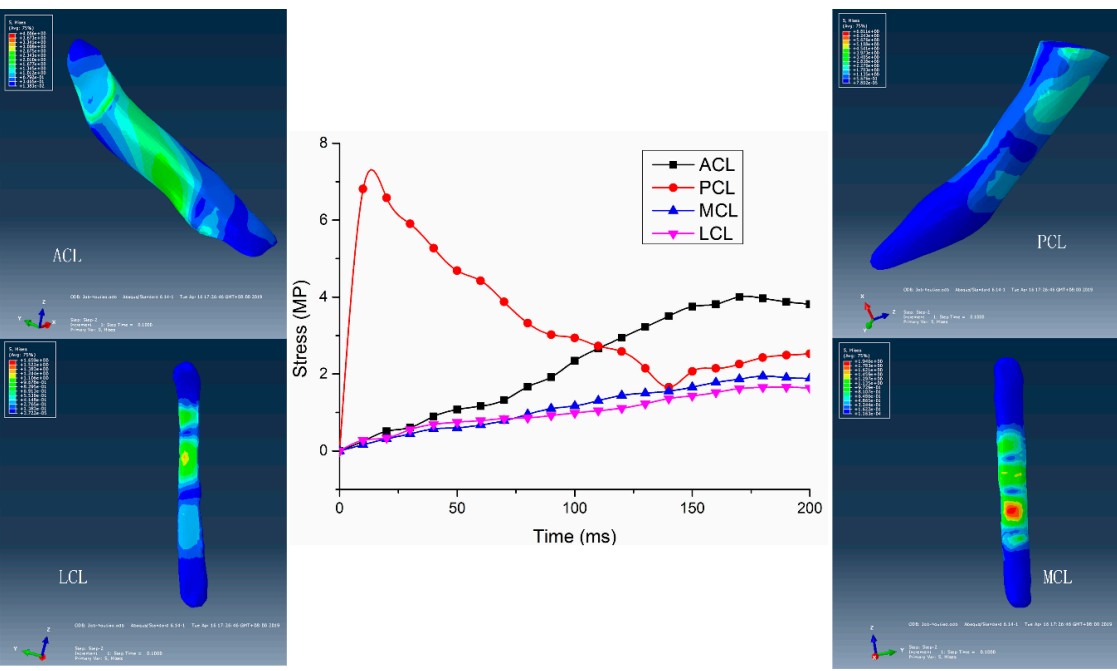

**Figure 7.** Stress distribution and stress-time curve of the ligaments in forward landing.

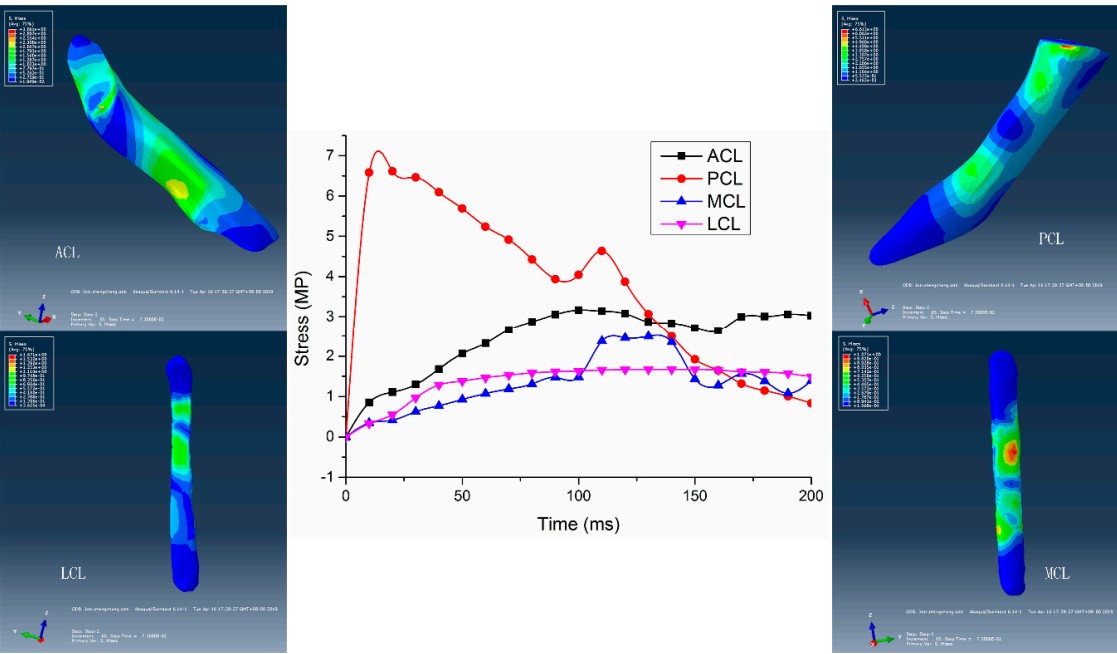

**Figure 8.** Stress distribution and stress–time curve of the ligaments in neutral landing.

During backward landing (Figure 9), elastic force from deformation of the tail of the skis provided an additional elastic force. Deformation of skis extended the duration of the balancing phase and reduced vertical impact force on the tibial plateau. According to the results of finite element analysis, maximum stress was experienced at the lower end of the posterior bundle of the ACL, at 4.52 MPa. On the PCL, stress was concentrated from the upper end of the posterior bundle to the lower end of anterior bundle, with a maximum value of 6.99 MPa. The middle of the MCL and LCL endured maximum stress, with values of 3.46 MPa and 1.39 MPa, respectively.

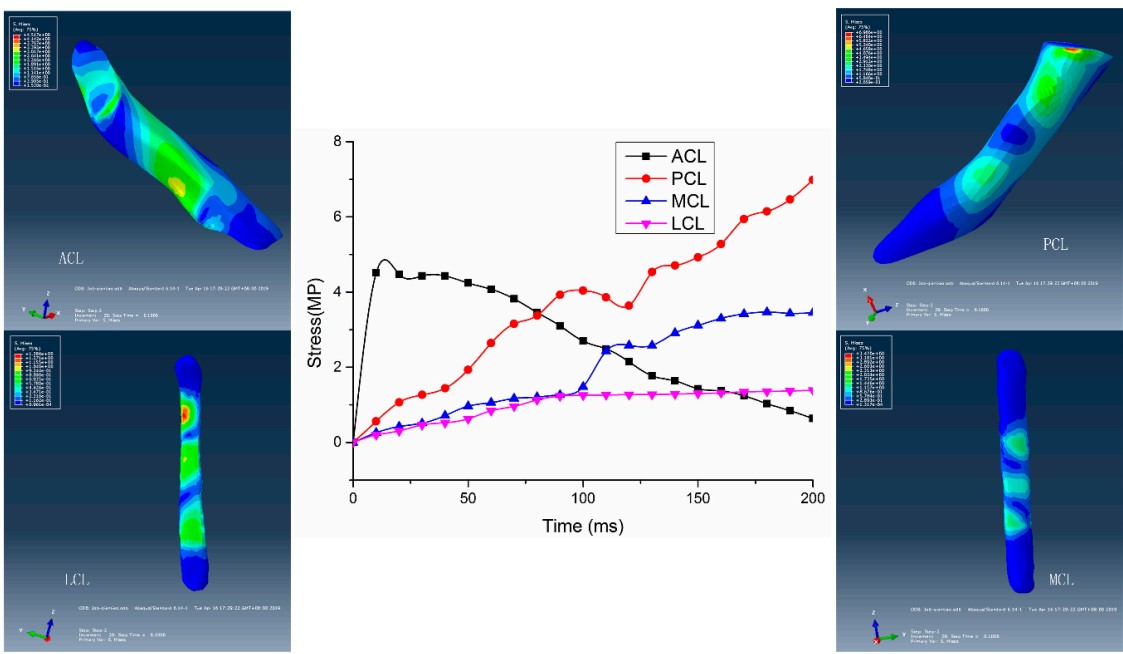

**Figure 9.** Distribution and stress–time curve of the ligaments in backward landing.

Simulation of stress at the tibial plateau are displayed in Figure 10, where the position of stress concentration was obvious. When the tibia moved rearwards relative to the femur, as in forward landing (Figure 10a), a region of stress concentration was observed at the front of the menisci and the middle of the medial meniscus. When the tibia is relatively immobile, such as during neutral landing (Figure 10b), the position of stress concentration was in the middle of the medial meniscus and the center of the lateral tibial cartilage. As the tibia moved forward relative to the femur such as in backward landing (Figure 10c), stress became concentrated in the middle of the medial meniscus and central position of the lateral tibial cartilage. Compared with neutral landing, the position and value of stress concentration in backward landing were relatively small and located towards the rear.

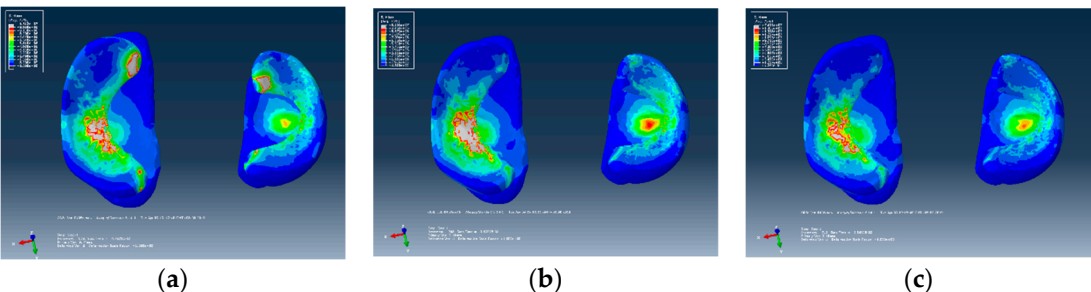

|   (a)   |   (b)   |   (c)   |

**Figure 10.** Stress distribution of tibial plateau and meniscus: (**a**) forward landing; (**b**) neutral landing; (**c**) backward landing.

## 4. Discussion and Implications

In the balancing phase of landing in freestyle skiing aerial sports, complex internal stresses are formed within the knee joint. In this study, only neutral, forward, and backward landing positions were considered. The principal differences between the three scenarios were position of center of gravity and the magnitude and direction of the reaction force.

In neutral landing, the reaction force provided by the landing slope was essentially perpendicular to the tibial plateau. Under the combined action of the impact force of falling and the frictional force of the snow, the analysis demonstrated that the tibia moved back and forth slightly within the knee joint. During knee flexion, the front and rear muscles of the thigh worked together to stabilize the knee joint.

According to the results of the simulation, peak stress on the tibial plateau and meniscus were greatest at this point in time for all three landing scenarios, but the relative displacement of the tibia was small. Stress on the cruciate ligaments in the balancing phase in neutral landing was the smallest of all three, and so was the safest form of landing.

In forward landing, slight backward motion of the tibia occurred within the knee joint and a vertical line through the center of gravity moved forward beyond the tibial plateau. As balancing progressed, the anterior thigh muscle was required to reposition the tibia [24]. According to the results of the simulation, peak stress in the tibial plateau and meniscus were smallest at this moment in all three conditions, with regions of stress concentrated in the anterior horn of the menisci. Because the medial meniscus was compressed by the femoral cartilage and resulted in lateral movement, stress in the middle of the MCL during the balancing phase of forward landing was the highest of all three landing scenarios.

In backward landing, elastic force of deformation of the skis and the impact force of falling resulted in the tibia exhibiting forward movement of the knee joint. As balancing progressed, the position of the tibia was adjusted towards the neutral position by action of the posterior thigh muscle [25] in a tangential direction, the tibia becoming pushed backwards. According to the results of the simulation, peak stress on the tibial plateau and meniscus was observed in forward and neutral landing, but of all three scenarios, stress on the ACL and PCL was largest in backward landing.

The results of the simulation in the three scenarios above demonstrate that the maximum value of stress (Table 1) in each ligament was less than the ultimate stress of the ligament. Even so, not all the ligaments are completely safe. Such as if athletes with the accumulated training, degenerative changes would occur within the region of stress concentration in the tibial cartilage and meniscus, in addition to fatigue and damage within the position of stress concentration in the cruciate ligament. If internal or external rotation of the knee joint occurred during landing, the risk of knee cartilage, meniscus or ligament injury would greatly increase. Over a controllable range, whether in forward or backward landing, aerialists require sufficiently strong anterior and posterior thigh muscles to maintain stability in the knee joint. If the strength of the anterior and posterior thigh muscles do not match, the risk of cruciate ligament injury is greatly increased.

**Table 1.** The stresses of the ligaments in three cases. (MPa).

| Type | Neutral Landing | Forward Landing | Backward Landing |
|------|-----------------|-----------------|------------------|
| ACL | 3.16 | 4.01 | 4.52 |
| PCL | 6.61 | 6.81 | 6.99 |
| MCL | 2.49 | 1.96 | 3.46 |
| LCL | 1.67 | 1.60 | 1.39 |

## 5. Conclusions

Although neutral landing had the greatest peak stress on the medial meniscus, stress on the ACL, PCL, MCL, and LCL were at their least of the three scenarios due to the slight relative displacement of the tibia, so neutral landing should be considered the safest and most successful technique of all three forms of landing.

During forward landing, a part of the femoral cartilage contacted with the front end of the meniscus, causing inward horizontal displacement of the meniscus. This allowed a greater likelihood that the medial meniscus would crush the MCL. The position of stress concentration in the medial meniscus was close to the anterior horn and middle edge, causing a high risk of meniscal damage. As the knee undergoes flexion, the tibia adjusted to a neutral position under action of the posterior thigh muscle. Therefore, athletes that have weakness in their posterior thigh muscle should avoid forward landing in order to reduce the possibility of PCL or MCL injury.

In backward landing, elastic deformation at the back of the skis created a cushion effect with deformation forces causing the tibia to move forward slightly. The tibia can be adjusted to a neutral

position only using the power of an athlete's anterior thigh muscles. Therefore, those aerialists that have weak anterior thigh muscles should avoid backward landings to reduce the risk of ACL injury.

No matter which landing condition is adopted, due to the forward velocity and rotational inertia of the lower limbs, under the combined action of the reaction force of the landing slope and friction between the skis and surface of the snow, shear force is evident on the tibial plateau with a trend of forward and backward displacement. In this study, internal and external rotations of the tibia were not considered. If tibial rotation occurs on landing, risk of cruciate ligament injury is greatly increased [26].

Peak stress on the tibial plateau and meniscus is extremely large. The medial meniscus is particularly prominent. Therefore, the design of professional knee protection should consider a reduction in landing impact with increased joint stability. Professional knee protectors can be effective in reducing the occurrence of sports injury and prolonging the career of athletes. The results of this simulation for three landing conditions provide a theoretical basis for the design of knee protection for freestyle skiing aerialists.

**Author Contributions:** Conceptualization, T.Y.; methodology, X.W.; formal analysis, Y.F.; writing—original draft preparation, Y.F.; writing—review and editing, T.Y. and X.W.; visualization, Y.F.; supervision, T.Y. and X.W.

**Funding:** This research was funded by the Key Special Project of the National Key Research and Development Program "Technical Winter Olympics" (2018YFF0300502) and the Natural Science Foundation Guide Project of Liaoning Province (2019-ZD-0517).

**Conflicts of Interest:** The authors declare no conflict of interest.

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
