# Peer review of "Simulation Analysis of Knee Ligaments in the Landing Phase of Freestyle Skiing Aerial"

_applsci, doi:10.3390/app9183713_

Round 1

Reviewer 1 Report

The work deals with a very interesting matter concerning the stress deriving from the landing phase of freestyle skiing aerial on the structures of human knee bones and ligaments. The work is very interesting and the methodology is accurate. In my opinion, the interesting application of different imaging from MR, CT etc. can be better described especially with reference to the mechanical parameters derived from RM and CT and their way to transfer it to FEM model. A little more extensive explanation on FEM model, constraints, load application etc. will complete the readability of the paper and clarify the aspects of the mechanical analysis. Some comparison with the stress produced by physiologic solicitation of the knee and some discussion about limit stress, namely elastic or strength stress level of the considered tissues would let the paper be more complete.

Reviewer 2 Report

The paper is well-written and interesting and deals with a good and interesting topic related to behaviour simulation analysis of knee ligaments in different landing scenarios. The study is based on complex work, both experimental and numerical.

I recommend some minor revisions:

Check the template

- References should be numbered in order of appearance and indicated by a numeral or numerals in square brackets;

- All figures and tables should be cited in the main text as Figure 1, Table 1… (Ex: line 98 – Figure 2 instead of Fig2, Fig. 6…….

At the end of introduction, is better to introduce the aim of paper. Line 48 - explain the notation CT and MR; Tables 1, 2, 3 – must rewritten in a proper format present in template; 2.4. – More information about what types of mechanical connections between femoral and tibia were chosen to simulate the knee joint? Where did you applied the constraints? The figures 2, 7, 8, 9 are not very clear. Line 141 – use space between ?,?and? Line 146 – use the measurement unit (N) for forces values In the discussion session please add a table summarizing your results (the maximum values of stresses for all cases of lending. Kind regards

Reviewer 3 Report

The authors presented a simulation analysis of knee ligaments in the landing phase of freestyle skiing aerial. The work was interesting and the results were instructive. I would like to recommend that it could be accepted in this journal after minor revision.

1) The format of the references need to be verified and revised.

2) Please give the full name of the abbreviation when it first appears.

3) Whether this neural landing is also the least damage to foot ring joint?

4) What are the limitations of this work?
